behaviour/evolution/ecology

segregation distortion, *t* complex, emigration, evolution of behaviour, *Mus musculus domesticus*, dispersal

**Author for correspondence:**
Jan-Niklas Runge
e-mail: jn.runge@protonmail.com

# Experiments confirm a dispersive phenotype associated with a natural gene drive system

## Jan-Niklas Runge and Anna K. Lindholm

Department of Evolutionary Biology and Environmental Studies, University of Zurich, Winterthurerstrasse 190, 8057 Zurich, Switzerland

(iD) J-NR, 0000-0002-0450-9897; AKL, 0000-0001-8460-9769

Meiotic drivers are genetic entities that increase their own probability of being transmitted to offspring, usually to the detriment of the rest of the organism, thus 'selfishly' increasing their fitness. In many meiotic drive systems, driver-carrying males are less successful in sperm competition, which occurs when females mate with multiple males in one oestrus cycle (polyandry). How do drivers respond to this selection? An observational study found that house mice carrying the *t* haplotype, a meiotic driver, are more likely to disperse from dense populations. This could help the *t* avoid detrimental sperm competition, because density is associated with the frequency of polyandry. However, no controlled experiments have been conducted to test these findings. Here, we confirm that carriers of the *t* haplotype are more dispersive, but we do not find this to depend on the local density. *t*-carriers with above-average body weight were particularly more likely to disperse than wild-type mice. *t*-carrying mice were also more explorative but not more active than wild-type mice. These results add experimental support to the previous observational finding that the *t* haplotype affects the dispersal phenotype in house mice, which supports the hypothesis that dispersal reduces the fitness costs of the *t*.

## 1. Introduction

Not all elements that make up the genome cooperate to increase the fitness of each other and of the individual that they produce [1]. For example, meiotic drivers manipulate the products of meiosis, usually sperm, to increase their own chance of transmission to the next generation, thereby decreasing the fitness of competing elements on the homologous chromosome [2]. Indeed, they often also decrease the fitness of the individual, thus only increasing their own. Hence, such elements are also called

selfish genetic elements. As a consequence of their 'selfishness', selfish genetic elements induce selection on the rest of the genome to suppress their activity. The typical suppression takes place on the level of the genome, either by the chromosome most negatively affected by the selfish genetic element or from elsewhere in the genome [3]. For example, in *Drosophila simulans*, *Dox* is a driver on the X chromosome that is suppressed by *Nmy*, an autosomal locus, in an ongoing arms race [4]. Suppression can also arise behaviourally, on the level of the individual, for example via mate choice favouring mates that suppress the selfish genetic element [5] or by increased female remating (polyandry), which decreases paternity success in carriers of selfish genetic elements that are poor in sperm competition [6]. How does the selfish genetic element respond to the selection imposed by such suppression?

We investigate this question using a selfish genetic element found in house mice *Mus musculus*, the *t* haplotype. The *t* haplotype is a variant of the proximal 35 Mb of house mouse chromosome 17 [7]. It is a supergene [8] consisting of at least four major inversions [7], which reduces the recombination rate to almost zero, thereby usually transmitting the *t* haplotype unchanged to the next generation [9]. Male heterozygous carriers of the *t* haplotype (notation: male $+/t$) transmit the *t* haplotype to over 90% of their offspring (instead of 50%), because *t* manipulates spermatogenesis in its favour [1,10], making it a meiotic driver. However, the *t* haplotype has major fitness drawbacks [11–16]. Mice homozygous for the *t* are not viable [17] or are sterile as males [18] and male $+/t$ perform very poorly in sperm competition, siring almost no offspring in matings with females that also mated with $+/+$ males [14,15,19], which is a stark contrast to monandrous matings, in which male $+/t$ transmit the *t* to almost all offspring due to their drive. Consequently, $+/t$ should be selected to increase the fraction of monandrous to polyandrous matings.

Recently, in a long-term observational study on free-living wild house mice, we found that juvenile carriers of the *t* haplotype, of both sexes, were more likely to emigrate [20], which is the first step of dispersal [21]. At average densities, $+/t$ were 48% more likely to emigrate. This difference further increased by 13 percentage points with a 46% (one standard deviation) increase in density.

The difference in dispersal could have evolved because *t* is expected to be much less fit in dense populations as females are more polyandrous at higher population densities [16,22]. Polyandry leads to sperm competition and controlled mating experiments show that a $+/t$ male sires only 11% of offspring when in sperm competition with a $+/+$ male [14]. Thus, any increase in the proportion of polyandrous litters impacts *t* fitness. In natural populations, Dean *et al.* [22] found that polyandrous litters were almost three times more common in high-density populations than in low-density ones. Density increases within a population also lead to increased polyandry in a phenotypically plastic response [16]. Such an increase in polyandry should greatly decrease *t*'s fitness [14,16,19], which was in fact observed in the long-term house mouse study [16], in which *t* eventually went locally extinct with increasing levels of polyandry.

Positive density-dependent polyandry also indicates a limited potential for males to fully dominate matings, which $+/t$ would need to do to sire offspring in dense populations. Another pressure that could select for $+/t$ dispersal comes from its homozygous costs, lethality or infertility, which translate into selection pressure to avoid $+/t \times +/t$ matings and thus high local *t* frequencies. *t*-frequency varies across populations and is higher in smaller populations [23], but there is no clear evidence that mice can detect *t* frequency [24–26]. In the light of this, a generally increased dispersal rate, also out of lower densities, could be selected. Therefore, while attempting dispersal is risky, we expect the difference in potential risks and rewards, when compared with not dispersing, to be shifted more towards taking the risk for $+/t$ than for $+/+$.

Currently, the link between dispersal and the *t* haplotype has only been found in an observational long-term study [20] and no controlled experiments have been conducted. Controlled experiments are, however, necessary to provide robust evidence, because they can be used to directly test the link between genotype, density and dispersal. Additionally, the previous study did not have the temporal resolution that an experiment can provide and as such was not able to measure differences in the timing of individual dispersal events. Delay until dispersal is, however, another important metric used to distinguish individual dispersal propensities in house mice [27] and more generally [28].

We extend our study of the *t* haplotype's impact on dispersal tendency by asking if an increased motivation to disperse is associated with a suite of other behavioural differences as part of a 'dispersal syndrome' [21,29]. Dispersal syndromes describe the values of traits that are found exclusively or predominantly in dispersing individuals. For instance, winged morphs are the predominant dispersers in locusts [30], and the *Pgi* polymorphism in butterflies is associated with greater dispersal, higher mobility and increased lifespan [31,32]. Fruit flies selected for increased dispersal show more locomotory and exploratory behaviour [33]. In mammals, dispersal polymorphisms are only known from mole rats,

which have an infrequent dispersal morph with increased body weight, changed hormone levels, increased locomotion, high motivation to disperse and increased interest in matings outside of their colony [29,34]. The links between dispersal and exploration and/or activity that have been found support the idea that a genotype that increases dispersal propensity could also modify the activity or exploratory behaviour to improve odds of successful dispersal.

Here, we test whether heterozygous $+/t$ mice differ from wild-type $+/+$ mice in their dispersal, activity and exploration phenotypes using replicated experimental set-ups, with a design adapted from previous studies [35,36]. Dispersal was tested in replicate enclosures with different densities of mice. We hypothesized that (i) $+/t$ would disperse more out of dense enclosures than $+/+$ in line with previous findings, (ii) $+/t$ would be more active than $+/+$ in a wheel-running test, and (iii) $+/t$ would be more exploratory than $+/+$ in an exploration experiment.

# 2. Methods

## 2.1. Study animals

The house mice *Mus musculus domesticus* used in this study were all laboratory-bred descendants from genetically wild mice caught in an intensively studied free-living population [36], in which dispersal has been previously studied [20]. Genetic drift in the laboratory population was prevented by regular additions of newly caught mice from the field study into the laboratory breeder pool. Thirty-four breeding pairs over 2 years were used to generate mice for this experiment, with 12 individual parents (17.6%) caught from the field study and the rest are $F_1$–$F_3$ laboratory-bred descendants of mice from the field study. $+/t$ and $+/+$ mice were derived from male $+/+ \times$ female $+/t$ crossings, in which approximately 50% of offspring are $+/t$. Additional $+/+$ mice were derived from $+/+ \times +/+$ crossings. $t$ haplotype status was determined by amplification of the *Hba-ps4* locus [37], as in [38].

Offspring were weaned at 23 days of age and were placed in same-sex same-age sibling cages. An experimental cohort of mice was selected from these to be 50% $+/t$ and also 50% female, equally distributed between the genotypes, while matched in ages as close as possible. Furthermore, we aimed to avoid the use of siblings within cohorts. In the high-density treatment, this was not entirely possible, so here all females were non-siblings and all males non-siblings. We achieved this by consistently selecting the founders to be eight brother–sister pairs to standardize conditions for both genotypes. Thus, the average relatedness was increased for both genotypes compared with the lower density. This could hypothetically confound the overall dispersal propensity, but we found in the previous long-term observational study that increased average relatedness only slightly decreased overall dispersal (S3 in [20]). Importantly, increasing relatedness did not affect the difference between the genotypes and also did not interact with the effect of the genotype on dispersal. Furthermore, mice are known to live in closely related groups [39], so the presence of siblings is not unusual.

When multiple individuals fulfilled the selection criteria, we selected an individual at random from a list of suitable candidates. We chose to use both males and females to create as natural an experimental set-up as possible, including reproductive competition. Genotypes and sexes did not differ in their age (adjusted $R^2 = -0.001$ of a linear model of age at enclosure begins with genotype and sex as predictors).

The average age at entering a cohort was 35.8 days of age (25 to 49 days, similar to the mice which dispersed in [20]). Each mouse chosen for a cohort was placed separately into a Macrolon Type II cage (267 × 207 × 140 mm) with ad libitum access to food (laboratory animal diet for mice, Provimi Kliba SA, Kaiseraugst, Switzerland) and water. The cages were outfitted with bedding (Lignocel Hygienic Animal Bedding, JRS, Rosenberg, Germany), kitchen paper, a cardboard tunnel (toilet roll) and cardboard pieces to provide hiding and nesting opportunities. We conducted first the activity, then the exploration experiments and then the dispersal experiment (see below for details of each experiment), cohort by cohort, with mice being weighed before each experiment.

## 2.2. Experimental set-ups

All experiments were conducted blind with regard to the genotype of the mice in so far as genotype information was only made available at the time of forming of cohorts and during analyses. Carrying the $t$ haplotype is not associated with any visible external phenotype.

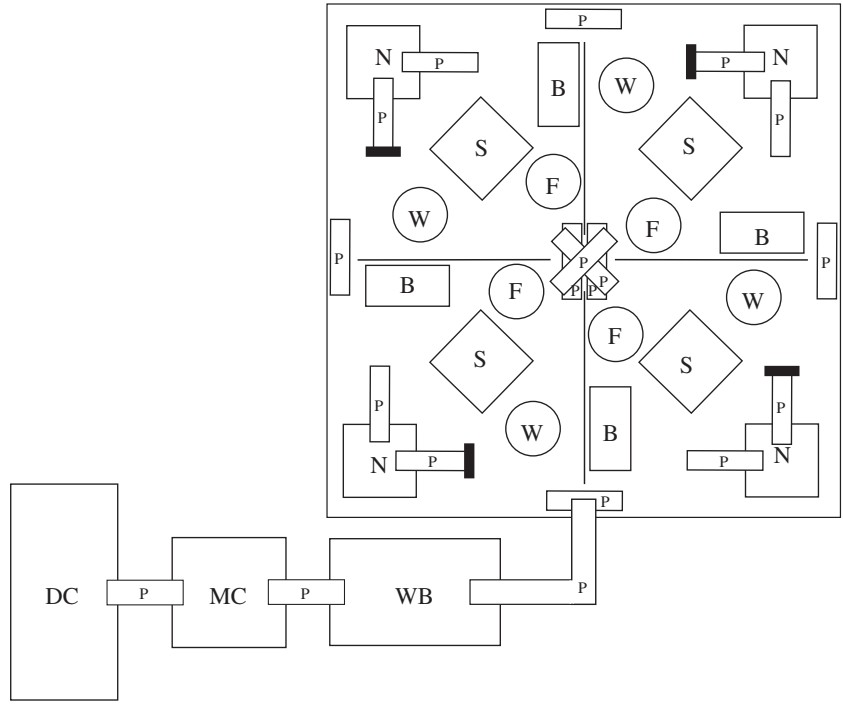

**Figure 1.** Schematic overview of an enclosure with a dispersal set-up (not to scale and only for illustrative purposes, see electronic supplementary material S1 for a photo). The enclosures contained four low walls that mice could cross and the following items: B: bricks with holes for shelter; MC: middle cage that only contained bedding; DC: dispersal cage in which the mice were found when they dispersed; F: food tray; N: nest-boxes with two pipes, but one of them was closed and only used for nesting; P: pipes that mice can move through; S: additional shelter made from a square tile angled against stones and/or sticks; W: water bottle; WB: water barrier through which the mice could swim to reach the dispersal cage.

### 2.2.1. Analysis

All plotting and analyses were performed in R 3.6.3 using the packages boot 1.3–24 [40,41], dplyr 0.8.5 [42], ggplot2 3.3.0 [43], lme4 1.1–23 [44], MASS 7.3–51.5 [45], pbkrtest 0.4–8.6 [46], readr 1.3.1 [46], readxl 1.3.1 [47], readODS 1.6.7 [48], reshape2 1.4.3 [49], Matrix 1.2–18 [50], car 3.0–7 [51], survival 3.1–12 [52], survminer 0.4.8 [53], coxed 0.3.3 [54] and AICcmodavg 2.3–1 [55]. The R code and its output can be found in the electronic supplementary material S2.

In all analyses, 'age' and 'weight' refer to the age at the start of each experiment and the latest measurement of body weight before the respective experiments.

### 2.3. Testing dispersal

We used 12 cohorts of either eight (low density, $n = 7$) or 16 (high density, $n = 5$) mice from our breeding laboratory (25% each sex/genotype combination), for a total of 136 mice. The low density (1.14 juvenile founders $m^{-2}$) was comparable to the mean minus one standard deviation density in the long-term study (0.80 juveniles $m^{-2}$), while the high density (2.29) was comparable to the mean plus one standard deviation in the long-term study (2.19). In the long-term study, densities were divided into adults and juveniles, with the juvenile density being the one that interacted with the genotype. These cohorts were implanted with a transponder for individual identification, and the following day were placed into 7 $m^2$ outdoor enclosures (figure 1), designed to resemble the environment in which mice from the long-term study analysed in [20] lived, but on a smaller scale, with the densities corresponding to below- and above-average numbers of juveniles $m^{-2}$ in the previous study. Similar enclosures are commonly used in studies of mice (e.g. [35,56,57]). This experiment had two purposes: to test for dispersal differences and to follow gene frequencies across generations (not reported here). Therefore, mice were allowed to reproduce freely, for an average of 107 ± 28 (s.d.) days before enclosure experiments were terminated. These experiments were conducted from April to December 2017 and June to November 2018.

We created an opportunity for the mice to leave the enclosures by swimming through a water barrier (figure 1; electronic supplementary material S1, figure S1) that was checked daily. This was designed based on previous studies that successfully used water as a barrier for mice that, when crossed, was called a dispersal event [35,58]. Hence, we will also refer to such events as dispersal in this paper. More details on the set-up can be found in the appendix.

### 2.3.1. Statistical analysis

The longest observed delay until dispersal was 53 days. A total of eight mice were found dead prior to 53 days from the start of the trial (3 $+/t$ males, 4 $+/+$ males, 1 $+/+$ female). We excluded these mice from the summarized dispersal percentages (such as in figure 3), because we did not count them as dispersers or non-dispersers. However, we included them in the survival analyses as censored data points, i.e. as not having dispersed until the day they died. We removed one non-dispersing $+/t$ female from all analyses because we could not reliably determine her fate due to the poor quality of her DNA sample and the loss of her transponder.

To analyse dispersal differences between $+/t$ and $+/+$ mice including the timing of dispersal, we built mixed-effects Cox models ($n = 135$) with philopatry (not dispersing) modelled as the response and dispersal as events with time as days since enclosure began. This is comparable to survival analyses that model survival as the response with death as events. We chose this approach because it allowed us to include the timing of the dispersal events and thus test whether a genotype dispersed more quickly. We controlled for the cohort, which shared an enclosure, as a random effect. Mice that never dispersed were censored to the number of days they spent in the enclosure. We used the *coxme* R package for this purpose. We evaluated whether genotype and the interaction of genotype and treatment density was informative for predicting dispersal by comparing models including these effects to models without using ANOVA comparisons with $\chi^2$-tests (with significance level $p = 0.05$). We evaluated whether breeding pair ID (the ID of the parent pair) improved the model when included as another random effect, which would imply a strong influence of genetic background or parental effects that should be controlled for.

In a second, exploratory, step, we aimed to build a null model of dispersal (excluding genotype) by backwards selecting from a full model (with predictors age, sex, weight and treatment), at each step excluding the variable with the highest $p$-value, with $p \geq 0.05$. We then tested whether a genotype and/or a genotype by treatment interaction improved that null model. Finally, we also tested whether any other genotype interaction would improve the model further.

Due to a lack of tools to plot and further analyse results of mixed-effect Cox models, we plotted the same results, but without the random effect, by copying the estimated coefficients and variances of the mixed model into a model without a random effect, effectively holding that effect constant. We created plots using *ggsurvplot* of the R package *survminer*. We calculated effect sizes using *coxed*, which bootstraps (10 000 repetitions) the data and provides confidence intervals for differences between two groups (here $+/t$ and $+/+$) based on Cox models.

We also analysed dispersal as a binary trait for comparability with the previous study, which ignores the timing of dispersal events. Details and an accompanying power analysis can be found in the electronic supplementary material S1. Furthermore, we also analysed males separately, but not females due to having few female dispersers, which interested readers can also find in the electronic supplementary material S1.

## 2.4. Activity test

Before entering the enclosures and the associated dispersal test, mice were given running wheels (Linton Instrumentation Ltd) in their solitary cages for three days as a measure of locomotor activity. The number and timing of revolutions of the running wheels were recorded (Columbus Instruments Device Interface 1.5 (Columbus, OH, USA)). The data of 53 mice, bred under identical conditions and selected using the same protocol, but which did not enter the dispersal experiment, were added to these analyses. Due to resource constraints, we first gave half a cohort, equally divided among genotypes and sexes, the running wheel for 72 h, then removed the running wheels, cleaned them with soap and water, dried them and placed the wheels with the other half of the cohort for 72 h. In total, 189 mice (25% each sex/ genotype combination) entered the wheel-running experiment, of which 11 were excluded from the analysis as they did not turn the wheel more than a few (16) times throughout the three nights, potentially indicating that they did not understand the apparatus.

**Figure 2.** Schematic overview of the set-up of the exploration experiment (not to scale and only for illustrative purposes, see electronic supplementary material S1 for a photo). Numbers indicate the five compartments that mice could explore in the experiment. 1 and 5: Macrolon Type III cages. 2 and 4: Tubes connecting cages. 3: Macrolon Type II cage in which the mouse (inside an enclosed cardboard tunnel) is placed at the start.

### 2.4.1. Statistical analysis

We recorded the revolutions of each wheel in 30 min windows (i.e. how many revolutions there were in 30 min). We first investigated the data visually by plotting the average wheel running per hour. We discovered a strong difference between active and inactive phases during the day. For later analyses, we only analysed the active hours, which were deduced to be from 21.00 to 06.00 (Central European Summer Time) based on the visual inspection. The active hours corresponded to the day/night cycle in the laboratory (dawn from 06.00 to 07.30, dusk from 19.30 to 20.30).

The least active individual had 191 revolutions and the most active had 50 238 revolutions (during the night). We created a linear mixed model with the response variable revolutions per hour, the random effects hour of the night (as a factor) and an individual ID. Similarly to before, we first tested whether cohort or breeding pair ID were useful random effects, with cohort now not included by default because the mice did not yet interact directly in this phase of the experiment. Then, we investigated the effect of genotype without covariates and subsequently again backwards selected models to find the best-fitting model in a second step, followed by an exploration of unhypothesized interactions. Due to the extreme distribution of activity with many individuals concentrated at the lower end, we used parametric bootstrapping model comparisons with the package *pbkrtest*, which provided the *p*-value, and described effect sizes with bootstrapping using the function *confint*, both methods with 10 000 repetitions.

## 2.5. Exploration

After the 6 days (2 × 72 h) of the running wheel experiment, 110 mice (26 +/+ males, 28 +/+ females, 28 +/*t* males, 28 +/*t* females) were placed into the exploration test, which was conducted during day time hours. The set-up consisted of a Macrolon Type II cage connected on two sides to larger Macrolon Type III cages (382 × 220 × 150 mm) by transparent tubes (figure 2; electronic supplementary material S1, figure S2 for a photo). For each iteration of the exploration experiment, a mouse was removed from its cage and placed into a container (590 × 390 × 420 mm), normally used to hold mice when cleaning cages. Inside this container, the mouse was introduced to a new cardboard tunnel, like the ones used in each cage for enrichment, without directly touching the mouse, to minimize anxiety [59]. The cardboard tunnel was placed directly in front of the mouse, with the side of the tunnel facing away from the mouse blocked by crumpled kitchen paper, until the mouse entered it. Upon entry, the side of the tunnel that the mouse entered from was also quickly blocked with a similar amount of crumpled kitchen paper. Then, the entire tunnel was enclosed by a sheet of kitchen paper that was modestly tightly wrapped and knotted. This was done with the paper itself, such that all materials (cardboard and kitchen paper) were easily breakable (e.g. by biting) and already known by the mice as these materials were used extensively in the breeding laboratory and the solo cages. Therefore, mice that were motivated to emerge from the tunnel should have been capable to do so. The tunnel with the mouse inside was then placed into the experiment. The set-up was closed with transparent plexiglas lids, with some space left on the sides for air flow, and the researchers left the room. A camera recorded movements of the mouse for 25 min, which was chosen to be longer than in a previous study on +/*t* exploration [60], because we gave the mice increased opportunity not to explore (by remaining in the tunnel). In a smaller pilot study, we found that this time frame resulted in variation from not exploring to quickly exploring the entire set-up and was hence deemed long enough to discover potential differences.

### 2.5.1. Statistical analysis

For each mouse, we extracted seven measurements from the video analyses (conducted with BORIS [61]): how many unique compartments the mouse visited in total (0–5), how often the mouse would move from one compartment to another per minute after the mouse emerged from the cardboard tunnel, and when it visited its first to fifth compartment for the first time. Every building block of the set-up was considered one compartment (two Macrolon Type III cages, one Macrolon Type II cage and two pipes, figure 2). A mouse was defined as having entered a compartment when all four legs were inside the compartment. There were two big groups of mice: those that never emerged from the tunnel ($n = 59$) and those that did ($n = 51$).

For the mice that emerged, we combined the highly correlated measurements into a PCA to generate an overall measurement of 'explorativeness'. Mice that did not enter all compartments were recorded as having entered the unentered compartments at minute 25 (the video length analysed per individual), e.g. a mouse that only explored three compartments, would be recorded as exploring the fourth and fifth compartment at minute 25. This was done to not have individuals with missing data. Similarly, mice that never emerged from the cardboard tunnel with more than 60 s remaining in the experiment were recorded as having had 0 movements between compartments per minute outside the tunnel. PC1 explained 79.08% of the variance in this dataset. It was strongly correlated (mean absolute $r$ of 0.89, lowest absolute $r$ was 0.79) with all measurements in the direction of increased exploration, meaning more compartments explored, more movements between compartments, and quicker exploration of compartments. Hence, we analysed PC1 as the response variable 'explorativeness'. We could not get a similarly clear association with PC2, which explained 10.43% of the variance, and have thus not analysed it further. We created linear models with PC1 as the response variable, once again first testing cohort, as the mice were sharing a laboratory room, and breeding pair ID as random effects using ANOVA comparisons. Afterwards, we tested the effect of genotype alone and then searched for the best-fitting model in an exploratory step. Because of likely violated linear model assumptions due to the rather flat distribution of PC1, we evaluated the 95% confidence intervals of the predictor coefficients generated by resampling the predictor values from 10 000 non-parametrically bootstrapped replicates using the function *boot*. We considered an effect to be statistically significant when the 95% confidence interval did not overlap 0 and removed the smallest non-significant effect during the backwards model selection process.

We also analysed whether emerging from the tunnel as a binary trait ($n = 110$) was explained by genotype and identified the best-fitting model using ANOVA comparisons with $\chi^2$-tests of generalized linear models.

## 2.6. Dispersal syndrome

Finally, to explore whether mice that dispersed differed behaviourally from non-dispersers in the exploration or activity experiment, we also built Cox mixed-effect models, controlling for the cohort as a random effect, with the predictors mean wheel revolutions per hour, median wheel revolutions per hour, explorativeness and emerging as a binary trait. We tested all of these separately in *ANOVA* comparisons.

# 3. Results

## 3.1. Dispersal

Twenty-four founders left their enclosure by crossing a water barrier, which we interpret as dispersal based on a previously established paradigm [35,58]. Of those 24, 16 were $+/t$ and 8 $+/+$ (figure 3), with 25% of $+/t$ dispersing in both low and high densities, while 20% of $+/+$ in low densities dispersed compared with 8% in high densities.

We built mixed-effect Cox models with the $n = 135$ founders (electronic supplementary material S1, tables S1 and S2) and first tested whether breeding pair ID (the parental pair) was a useful random effect but did not find this to be the case ($p = 0.98$, $n = 135$). Then, we tested our hypothesis by evaluating whether a model with genotype, treatment density and their interaction was better than a model with no predictor, but did not find a significant difference ($p = 0.14$). However, only adding genotype without treatment density did improve the model ($p = 0.049$).

In a subsequent step aimed at controlling for potentially confounding variables, we built a model with all non-genotype effects and backwards selected a null model that only contained significant

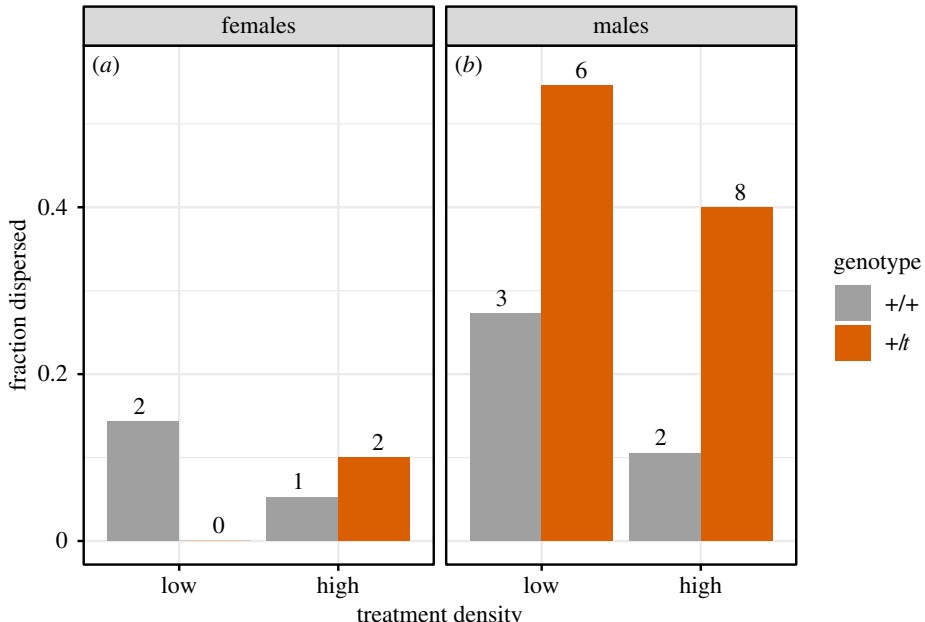

**Figure 3.** Proportions of dispersers of the two genotypes in the two treatment densities, divided by sex ($n = 127$). Numbers indicate the absolute counts of dispersers in each category.

effects, which left us with only sex as an explanatory variable (removing age at enclosure start ($p = 0.84$), then treatment density ($p = 0.41$), body weight before the enclosure ($p = 0.16$) and keeping sex ($p = 0.001$)). Adding genotype also improved that null model ($p = 0.03$), but again the treatment interaction did not improve it further ($p = 0.45$).

Finally, we explored the data for unhypothesized interactions. We tested whether any interaction of genotype with another variable would improve that model and found a significant interaction between genotype and body weight that predicted dispersal ($p = 0.006$), but no other interaction (sex: $p = 0.10$, age: $p = 0.36$). Hence, the most informative model included genotype, sex, weight and interaction between genotype and weight. Given this model, 1 g heavier than average $+/t$ dispersed 27.81 days (95% CI: 4.48 to 62.89 days) earlier than 1 g heavier than average $+/+$, but 1 g lighter than average $+/t$ dispersed only 10.66 days earlier ($-15.48$ to 55.66) than 1 g lighter than average $+/+$ (figure 4). Importantly, $+/t$ mice did not differ in their weight (before the enclosure) from $+/+$ mice when controlling for breeding pair ID as a random effect (electronic supplementary material S1, tables S3 and S4, 95% CI of $+/t$ effect: $-0.08$ to 1.31 g), but males differed from females (3.56 to 4.87 g) and age increased weight (0.12 to 0.26 g per day). Mice did also not differ in weight with regard to density treatment ($-0.17$ to 1.39) or treatment interacting with genotype ($-0.83$ to 2.19).

## 3.2. Activity

We tested for differences in locomotor activity using a set-up in which each mouse had access to a running wheel for 72 consecutive hours, but we only analysed the night hours (in which the mice were most active). Breeding pair ID ($p = 0.04$), but not cohort ($p = 1.00$), did improve model fit and was hence controlled for as a random effect. $+/t$ mice did not differ significantly from $+/+$ mice in their wheel revolutions per hour (electronic supplementary material S1, tables S5 and S6; figure 5), without covariates: $p = 0.84$, $n = 178$. The only significant term remaining after backwards selection of the other variables was sex, with males on average running 372.45 (CI: $-469.02$ to $-275.27$) fewer wheel revolutions per hour than females ($p = 0.0001$). Age and weight did not have a significant influence ($p = 0.94$ and $p = 0.15$, respectively). Genotype did not improve a model that controlled for sex either ($p = 0.71$) and similarly, no interaction with genotype of any of the other variables (sex, age and weight) was significant ($p = 0.25$, $p = 0.48$ and $p = 0.51$, respectively).

**Figure 4.** Survival (here, philopatry) curves (with 95% confidence interval) for $+/t$ and $+/+$ based on model predictions ($n = 135$) for either 1 g heavier or lighter than average males.

## 3.3. Exploration

We scored the behaviour of $+/+$ and $+/t$ in an exploration set-up and extracted the first principal component of a PCA of all scored exploration behaviours ('explorativeness', figure 6 for genotype distributions in three of the scored variables). However, we only analysed explorativeness in mice that emerged (28 $+/t$ and 23 $+/+$), rather than in all mice, which would have included mice that never left the tunnel in which we placed them into the set-up (53.6% of the sample, $n = 110$). Explorativeness values (measured as PC1, thus with a mean approximating 0) ranged from −3.21 to +4.86 (figure 7, $n = 51$). Explorativeness (PC1) correlated positively with the number of unique compartments explored ($r = 0.79$) and the compartment changes per minute ($r = 0.86$), and correlated negatively with the delay until a mouse explored the first to fifth unique compartment ($r = −0.88$, $r = −0.94$, $r = −0.94$, $r = −0.93$, $r = −0.88$, respectively).

Cohort and breeding pair ID did not improve the model of explorativeness and were subsequently ignored (electronic supplementary material S1, tables S7–S9, $p = 0.41$ and $p = 0.74$, respectively). Bootstrapped predictors of a linear model of explorativeness explained by genotype found that $+/t$ had a PC1 that was increased (more explorative) by 1.36 (95% CI: 0.16 to 2.59) over $+/+$'s PC1. For interpretation, this translates into 231.91 s (−303.48 to −160.34) earlier emergence or 0.65 (0.36 to 0.94) more compartments explored based on regression of these variables against PC1. Since the confidence interval did not overlap 0, we considered this effect to be significant. By contrast, sex, age and weight

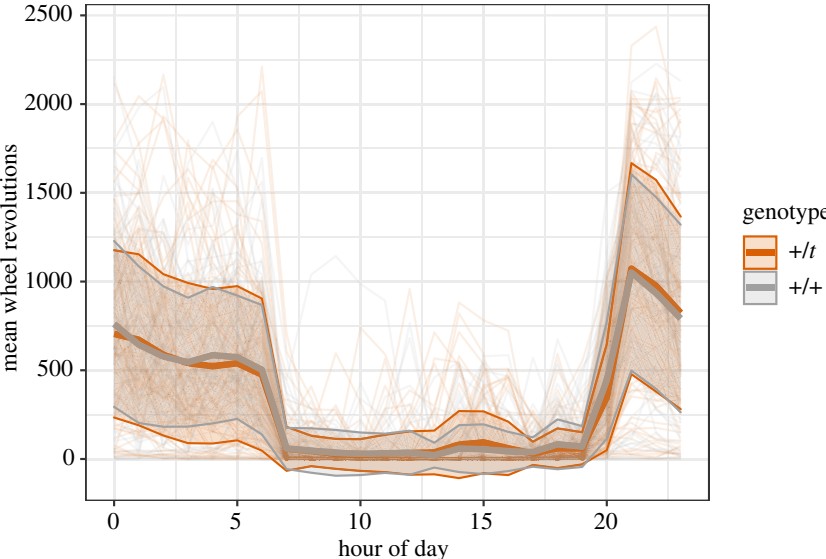

**Figure 5.** Average wheel revolutions (bold lines, standard deviations in shaded areas) per genotype over the course of a day. Individual ($n = 178$) averages are plotted in translucent, smaller lines.

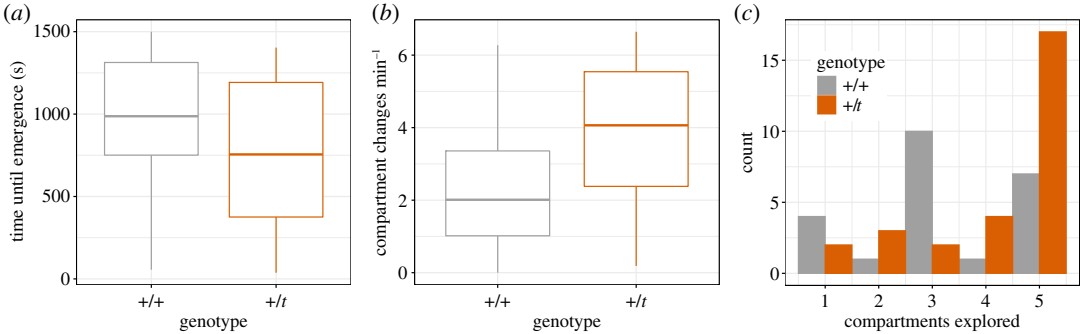

**Figure 6.** Distributions of the genotypes in three behavioural traits correlating with PC1 'explorativeness' ($n = 51$): (a) time until the mouse emerged from the tunnel, (b) the number of changes between compartments per minute after leaving the tunnel, and (c) the number of unique compartments the mice explored. Boxplots show the median as a line, first and third quartile as hinges, and the largest and smallest values as whiskers.

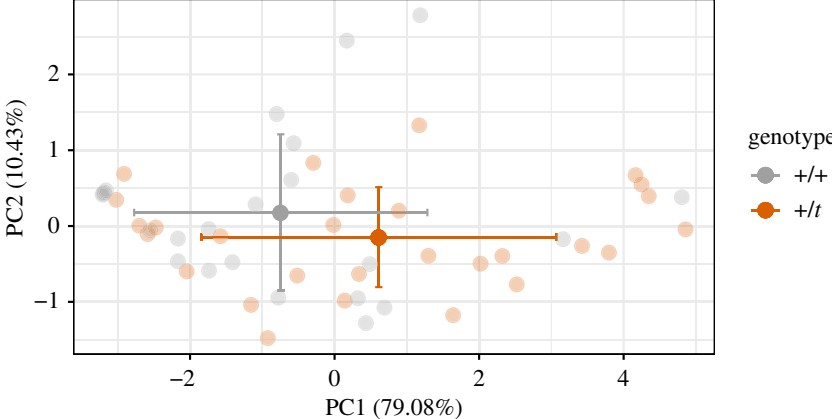

**Figure 7.** Individual ($n = 51$) PC1 and PC2 values of a PCA of exploration set-up variables of the mice that left the tube and started exploring: how many unique compartments the mouse visited in total (0–5), how often the mouse would move from one compartment to another per minute since the mouse emerged from the tunnel, and when it visited its first to fifth compartment for the first time. The means of both genotypes are shown in non-translucent dots with the standard deviations as error bars.

did not explain explorativeness, as their effects always overlapped 0 during backwards model selection (e.g. in a model with all three: −2.92 to 1.05 (males), −0.2 to 0.16 (age) and −0.27 to 0.42 (weight)). Furthermore, neither of these traits interacted with the genotype to explain explorativeness (males: −2.18 to 2.93, age: −0.2 to 0.34, weight: −0.45 to 0.27).

Twenty-eight $+/t$ and 31 $+/+$ never emerged from the tunnel in which they were placed into the exploration set-up during the trial, and hence never started exploring. A model including genotype as a predictor did not explain emergence better than a model without predictors (electronic supplementary material S1, tables S10 and S11, $p = 0.44$, $n = 110$). Via backwards selection, a null model was found to contain sex ($p = 0.01$, male over female odds: 2.66 (1.22 to 6.00)) and age ($p = 0.04$, odds increase per day: 1.08 (1.00 to 1.16)), but not weight ($p = 0.60$). Genotype did not improve this null model either ($p = 0.28$) and no interaction was significant (sex: $p = 0.56$, age: $p = 0.46$, weight: $p = 0.71$).

## 3.4. Mice that emerged were more likely to disperse

We found that the only behavioural trait that predicted dispersal from all the possible combinations was whether mice emerged (as a binary trait, $p = 0.04$, $n = 108$; electronic supplementary material S1, tables S12 and S13), but not how much they explored if they explored ('explorativeness', $p = 0.83$, $n = 50$), or either median or mean wheel-running activity ($p = 1.0$, $p = 0.68$, $n = 127$). Mice that emerged dispersed 17.51 days earlier (0.58 to 42.25) than mice that did not.

# 4. Discussion

We confirmed that carriers of the $t$ haplotype are more dispersive than wild-type mice using an established experimental approach in which mice departing from a population by crossing a water barrier are defined as dispersers [35,58]. The experimental approach used here provides direct support for the hypothesis that the $t$ haplotype affects dispersal, with twice as many $+/t$ than $+/+$ dispersers. This relationship was first observed in a long-term study of wild house mice [20], but remained untested in a controlled environment until now. The $t$ haplotype is the first example of a selfish genetic element impacting dispersal and adds to a small number of known dispersal polymorphisms. This finding could explain in part how the $t$ haplotype has persisted despite its fitness drawbacks and potentially how it avoided counter-adaptations by the rest of the genome in an increasingly complex arms race [5].

In contrast with the long-term study, we did not find clear evidence that the difference in dispersal between $+/t$ and $+/+$ increases in higher densities. Here, we found that the difference in the fraction of $+/t$ versus $+/+$ that dispersed was higher in higher densities, but this difference was not statistically significant. Our study was sufficiently powered to detect an effect of the same magnitude as we observed in the long-term study (see electronic supplementary material S1). However, dispersal rates were overall lower than in the long-term study, which decreased the likelihood of detecting differences. We designed the dispersal experiment to be as similar as possible to the original study, including the age of the mice and the density of mice of that age. However, the absolute number of mice was higher in the long-term study population than in the experimental enclosures. Depending on how mice gauge density, the experimental conditions may have not provided signals that would lead the mice to interpret a condition of high density. Compared with house mouse populations more generally, the densities we used were more on the average side (1.14 and 2.29 mice m$^{-2}$ compared with the reported 0.01 [62] or 7.06 [39]). Higher densities could be considered in future studies, but would be logistically demanding.

In general, males were much more dispersive than females, which is not what we found in the long-term study, in which females were a bit more dispersive than males [20], but both effects have been found in house mice (reviewed in [63]).

An alternative interpretation to innate differences in dispersal propensity may be that wild-type mice could force $t$-carrying mice to disperse, but there is mixed evidence on whether mice can even detect if another mouse carries the $t$ haplotype [24], with no evidence found in our population [25,26]. Additionally, there is also no clear evidence on whether $+/t$ mice are more or less dominant than $+/+$ mice [12,64,65], and there was no difference in the fraction of dead juveniles for each genotype in our previous field study [20]. Consequently, a difference in dispersal propensity is the simplest explanation for the pattern found so far.

Consistent with our past observational study [20], we did not find a difference in the effect of the genotype between the sexes. The interest of the *t* and a male carrying it are aligned when it comes to dispersal due to polyandry, because polyandry harms both. Hence, one could assume dispersal to be increased in +/*t* males over +/*t* females. However, one generation later, +/*t* females that dispersed could also improve the fitness of their +/*t* sons, which is even more so the case from the *t*'s, rather than the individual's perspective. As such, there may not be a selection towards sex bias in *t*'s dispersal effect, which would be consistent with our results.

Surprisingly, however, we found that heavier +/*t* mice were more likely to disperse than heavier +/+. We did not hypothesize this *a priori* and so we consider this an exploratory result. We did not find an effect of body weight in the previous study [20], in which we were only able to test for an impact of the body weight as a pup rather than weight as a sub-adult as we did here. In house mice, male dominance status is positively predicted by body weight [66] and subordinate males are evicted by dominant males and are thus more likely to disperse [67,68]. This is consistent with the pattern we found in +/+, but not with the one in +/*t*.

Studies on other species found more positive than negative effects of body weight on dispersal propensity [69], but both are common [70,71]. The direction of the effect can depend on the genotype [72]. In naked mole rats, distinct dispersal morphs are bigger than their resident conspecifics [34], possibly to mitigate costs of dispersal [73]. From our results, it appears that in house mice the effect of body weight could depend on whether a mouse carries the *t* genotype: +/*t* could increase the odds of successful dispersal if they remain prone to disperse at higher body weights while heavier +/+ may be rewarded with increased chances of successfully establishing as dominant residents if they do not disperse. Further work on this novel and unpredicted result will be needed.

We found that +/*t* showed more explorativeness than +/+, but only after they emerged from the tunnel. On the other hand, the genotype was not predictive of this emergence, which has been interpreted as an anxiety measure [35]. Exploration is one of the behaviours hypothesized to be part of dispersal syndromes [29]. Dispersal events could be preceded by bouts of exploration, in which the disperser-to-be searches for new territories, as is the case in deer [74] and squirrels [75]. However, a previous study on adult +/*t* under different experimental conditions did not find a statistically significant difference in exploratory behaviour with the mean being elevated in +/*t*, but overlapping +/+ in its distribution [60]. Combined with our results, this could suggest that behavioural differences of +/*t* and +/+ that are connected to dispersal may be found primarily in juveniles or sub-adults, which are the primary dispersers in house mice [62].

Activity levels in the wheel-running experiment very clearly did not differ between +/+ and +/*t*. This is in contrast with our hypothesis. However, previous experiments on adult mice, using an empty cage rather than running wheels, found that adult +/*t* females were less active than their +/+ conspecifics [60], which lined up with differences in food consumption [60], resting metabolic rates [76] and lifespan [19]. We speculate again that the difference in age between the mice in our experiment and the mice of previous studies could play a role. Females were found to turn the running wheels much more often than males, which has been found by others as well [77,78].

Interestingly, we found that whether a mouse emerged from the tunnel in the exploration set-up (and started exploring) positively predicted dispersal. By contrast, what happened afterwards in the exploration set-up and the wheel-running activity did not predict it. This result is at odds with the fact that +/*t* have increased dispersal and increased explorativeness, but not increased the probability of emerging. Additionally, a previous study using enclosures with a dispersal opportunity and compartment exploration behaviour tests found exploration activity rather than emergence timing to predict dispersal [35]. To discover a potential confounding effect of the social environment, it may also be helpful to consider exploration experiments in a social context and/or dispersal experiments without one. In summary, the relationship between behaviours related to exploration, dispersal in the enclosure, and genotype is not yet clear and requires additional investigation.

## 5. Conclusion

Carrying the *t* haplotype increased the propensity of mice to explore and disperse. Its carriers were more prone to disperse at higher body weights than +/+. Taken together, the *t* haplotype affects the dispersal phenotype of house mice. It is important to remember that the *t* haplotype is also expected to spread very rapidly in populations of low density and low *t* frequency due to the maximized effect of its drive and minimized impact of its deleterious traits (poor sperm competition and lethal homozygosity). Thus, *t* is

not just more likely to disperse than the rest of the genome, but $+/t$ should also be particularly successful as immigrants into a population in which $t$'s drive is stronger than its disadvantages. However, so far nothing is known about whether $+/t$ are particularly well-adapted behaviourally to be immigrants, which would inform us about differences in dispersal risks for $+/t$, whereas our argument relies on differences in benefits of dispersal.

In summary, the $t$ affects the dispersal phenotype of its carrier. Dispersal may reduce the negative fitness effects of the $t$ haplotype and increase the expression of its advantageous trait, drive, which, all in all, supports the hypothesis that the increased dispersal of $+/t$ is selected directly rather than a by-product (see [79]). This hypothesis can also be tested in other meiotic drive systems with similar fitness (dis-)advantages, such as $SR$ in $Drosophila$ [6]. Our results also motivate additional studies on dispersal-related traits in $+/t$, such as selfishness [80].

Ethics. The experiments were carried out under the permits ZH075/18 and ZH134/16 of the cantonal authorities in Zurich, Switzerland. The aim of the study was to create a naturalistic environment for the mice. To avoid causing many artificial disturbances potentially leading to researcher-induced dispersal events, we monitored the enclosures every weekday and 1 day on the weekend without opening nest-boxes and only thoroughly monitored every 10 days while searching for litters. On thorough searches, we made sure to see every mouse in an enclosure and our protocol was to remove mice that were in poor condition. We provided plenty of opportunities for fleeing, hiding and, importantly, leaving the enclosure via dispersal. All animals were sacrificed at the end of the experiment.

Data accessibility. The dataset generated and analysed during the current study is available in the Zenodo repository under https://doi.org/10.5281/zenodo.3865626 [81].

Authors' contributions. J.-N.R. and A.K.L. conceived the study and wrote the manuscript. J.-N.R. analysed the data and conducted the majority of the experimental work (see Acknowledgements). All authors read and approved the final manuscript.

Competing interests. The authors declare that they have no competing interests.

Funding. This study was funded by the Swiss National Science Foundation (31003A_160328) and the Claraz Stiftung.

Acknowledgements. We thank Jonas Cheung and Jasmine Klasen for scoring exploration videos. We further thank Aline Ullmann, Vishvak Kannan, Lennart Winkler and David Hug for their help in conducting the experiments. Furthermore, we thank Laura Lüthy and Seija-Mari Filli for their help in conducting the exploration and running wheel pilot study. We are grateful for the work of Marcel Freund in building the water cages. We thank Bruce Boatman for his support in running the experiment as well as him and Barbara Schnüriger for their work in the mouse breeding laboratory. Finally, we thank Jari Garbely for his work in DNA extraction and identification of $t$ haplotype status.

# Appendix A.
## The enclosures

The enclosures consisted of concrete floors, upon which we placed bedding material, the same we used in the cages, and walls to prevent mice from climbing out. The enclosures were themselves caged to protect against predation and protected from the rain with a Plexiglas roof and from the sun by a tarp hanging overhead. Each enclosure had four nest-boxes, with one entry tube, representing high-quality nesting sites, one in each corner, and four low dividing walls arranged like a plus-sign with space in the middle of the enclosure and on the ends to allow for movements while also facilitating territory formation. We used tubes, bricks, sticks, tiles, stones, kitchen paper, straw and cardboard rolls to provide hiding and evasion opportunities, nesting material and enrichment. We had four feeding sites to which the mice had ad libitum access, filled with a half-and-half mix of hamster food (VITA-BALANCE 26267 by Landi AG, Switzerland) and oats, the same as used in [36], including in the emigration study of [20]. We also had four drinking sites per enclosure. At the start of an enclosure, the solo cages were placed without their lids into the enclosure, allowing mice to enter the enclosure without handling them directly.

Mice in the enclosures were able to leave via a tube that led out of the enclosure and towards an enclosed plastic box ($290 \times 200 \times 220$ cm) filled with about 8 cm deep water (called the 'water cage') with a tube on the other side leading to a Macrolon Type II cage filled only with bedding (used for occasional camera trapping, but kept for all experiments), leading via a tube to a Macrolon Type III cage (called the 'dispersal cage') with bedding, cardboard, food and water, modelled after what is used in our breeding laboratory and intended to be attractive. Both tubes leading in and out of the water were placed circa 4.5 cm above the water with ladders (steel sheets with holes) covering the distance between tube and water. There was a divider from the top down to circa 11 cm above the ground in the middle of each water cage to prevent the mice from

jumping over the water. The access to this dispersal opportunity was ensured in the first two weeks of an experiment (excluding the weekend and/or short breaks for cleaning of the apparatus). Afterwards, access was recurring but not constant, because mice started nesting in the tube leading to the water, which presumably would have interfered with dispersal attempts. Removing constant access was intended to discourage this behaviour and whenever nesting occurred, the dispersal cage access was blocked off and the apparatus cleaned. Sixty-seven per cent of dispersal took place in the first two weeks and 83% in the first three weeks, similarly to [35] where most dispersal took place within 7 days. Thus, we assume that we found a good balance between continuous access to the water cage in terms of time and making sure that the water cage was actually accessible for all mice by having it clean and without nests, i.e. outside of territories. Each morning on weekdays (with access to the apparatus being blocked on weekends), the dispersal cage was checked for the presence of mice. Upon finding a mouse, the cage was removed with the mouse inside and the mouse was declared a disperser (and would not return to the enclosure). The dispersal apparatus was subsequently cleaned with soap and water and set up again.

Mice that founded the enclosures as well as mice that were born in the enclosures were able to disperse. Births in the enclosures were monitored with regular searches for new offspring, every 10 days. These offspring were tissue sampled for genetic analysis at 13 days of age and returned to the nests. To our surprise, we only ever recorded 2 out of over 500 offspring dispersing from the analysed enclosures. Hence, we did not investigate that data further for this purpose and instead focused on the founders which dispersed much more often.

We decided against preventing the mice from leaving the dispersal cage once they entered it, because it would make it harder to interpret whether they actually wanted to leave the enclosure if they had no way of returning. We used video cameras to monitor the behaviour of mice entering a portion of the dispersal cages. Out of 96 filmed nights, eight mice were seen on the dispersal side of the water barrier, with five of these found by us the next morning (and thus recorded as dispersers) and three returned to the enclosure before we could find them. Out of the three returning mice, two returned to the enclosure quickly (within roughly an hour), indicating an exploratory trip rather than dispersal and one stayed until the morning, just short of being found by us. Furthermore, in 96.7% (86/92) of days that we did not find a disperser, there was also no mouse in the dispersal cage at any point. Considering that mice that stayed overnight in the dispersal cage were found in five out of six times and the highly accurate call of nights without dispersal, we assume our measurement to be accurate enough for further analysis.

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
