## [Peer Review File · Royal Society Open Science]

Review History

RSOS-202050.R0 (Original submission)

Review form: Reviewer 1

Is the manuscript scientifically sound in its present form?

Yes

Are the interpretations and conclusions justified by the results?

No

Is the language acceptable?

No

Do you have any ethical concerns with this paper?

No

Have you any concerns about statistical analyses in this paper?

No

Recommendation?

Accept with minor revision (please list in comments)

Comments to the Author(s)

See the attached document (Appendix A).

Review form: Reviewer 2**Is the manuscript scientifically sound in its present form?**

Yes

Are the interpretations and conclusions justified by the results?

Yes

Is the language acceptable?

No

Do you have any ethical concerns with this paper?

No

Have you any concerns about statistical analyses in this paper?

Yes

Recommendation?

Accept with minor revision (please list in comments)

Comments to the Author(s)

This study provides experimental support for the hypothesis that t carriers in house mice are more likely to disperse. In addition, the study includes two analyses of behaviors (in +/+ and +/t mice) related to dispersal, including overall activity level and exploration. While the t haplotype did not contribute to differences in activity level, it did explain variation in some aspects of exploratory behavior. Understanding how the t haplotype influences behavior may help identify how it spreads despite significant negative effects on fitness related to sperm competition and homozygous inviability. The authors interpret their results in this context.

The experiments were generally carefully designed and the analysis clear and motivated. The experimental enclosure is unique and challenging to execute. The results have significance for both the broader research area of selfish genetic elements as well as evolutionary genetics in house mice.

Major Comments

1. Provide more context on the previous observational study to motivate this work and to lay the groundwork for interpretation of the results in the discussion.
2. The potential link between dispersal and selection against t carriers due to t/t inviability is easy for the reader to understand. The link between population density and mating multiply and selection against t carriers needs to be developed further in the abstract (briefly) and introduction (more room here to discuss). For example, to what extent does mating multiply (and selection against t carriers) depend on density? Is there a strong relationship between density and number of matings? Does the disadvantage to t carriers occur as a threshold (i.e. any more than one mating is very bad for t carriers) or is it linear (increasingly detrimental as the number of matings increase)? Does data suggest that female mice mate with at least two males in even less dense populations? Some studies are cited, but they need to be discussed.

3. There are some experimental design considerations (variation in relatedness among mice) and questions of statistical power/approach that should be addressed more directly in the text. Specific examples are given in the detailed comments.
4. There is an opportunity to strengthen this manuscript by revising the Discussion. This section is less organized and clear and the tone less formal and more speculative than the rest of the manuscript.

Detailed Comments:

1. Line 13- It would be helpful to note the type of study that produced the previous result (observational) as a contrast to this controlled, experimental approach. How strong was the evidence from the observational study?
2. Line 17- It would be good to re-phrase "heavier t-carrier" a bit. No mention of weight or body size has been made before this and it is not clear what "heavier" means. Is it weight per unit length? Is it controlled for age?
3. Line 20 - this is a little speculative. I think it is more well-supported to say that the results provide experimental evidence that the t haplotype affects dispersal and adds support to the hypothesis that dispersal may reduce the fitness costs of the t.
4. Line 33 -Explain a little bit more about genetic mechanisms of drive suppression to contrast with the work here on behavior. Readers who are less familiar with genetic drive could use more context.
5. Line 43- Give the chromosome number
6. Line 54- "Recently, in a long-term observational study on free-living house mice, we found that". There is also room here to elaborate on the results and the strength of the conclusions to better motivate this experimental approach.
7. Line 58- This is a good place to put in more detail about the association between density and number of mates per estrous.
8. Line 59- I think "positively" should be "positive"
9. Line 69 - Why are controlled experiments necessary? E.g. "Controlled, manipulative experiments can be used to directly test the link between the t-haplotype and dispersal at different densities."
10. Line 71 - I don't think you need this sentence here. It seems like something better for the discussion where you can bring together the results of the two studies and discuss the evidence for the hypothesis.
11. Line 74- This sentence is missing mention of the t haplotype.
12. Line 98- Can you provide some details on the genetics of the colony -i.e. how are individuals related to each other or is there a breeding scheme that can give context?
13. Line 109- Higher relatedness among the mice in the high-density treatment -i.e. males and females from the same parents seems like an experimental design problem. I appreciate that generating the mice for the experiment is very time consuming, but it is plausible that relatedness could be a confounding factor. It would be helpful to discuss the extent of the issue - e.g. if very few brother-sisters were in the same cohort than the impact could be trivial. It would also be good to talk about the natural history of house mice and what might already be known about the impact of relatedness on dispersal to help address this concern.
14. Line 121- This is subjective, but I would give the methods and results in the order of the tests. It was a bit confusing to have the enclosure and then the other two tests.
15. Line 135- were lengths measured as well?
16. Line 137 - Discuss the choice of these densities in the context of house mouse natural history rather than just the previous study.
17. Line 146 - "an average of 107.."
18. Line 154- I like the photograph, but it may be better to put this in the supplement and use a diagram here instead, so that the reader better understands the dispersal barrier as a water cage. It is difficult to see that with the plastic bin that is pretty opaque. The external cages look haphazardly set up.

19. Line 160 – I did not understand this sentence.
20. Line 162- Explain what “censored data points” are for the reader.
21. Line 166- This analysis is critical for the reader and I think it could be clearer. Explain what philopatry is in this context and why you are analyzing dispersal using a survival model type approach as opposed to alternatives. For example, you do test emergence from the tunnel later as a binary trait, did you test dispersal in the enclosures as a binary trait?
22. Line 271 – Had the mice been “cohorted” at this point?
23. Line 322- Given the very low rate of female dispersal and the low absolute number of female dispersal events, did you consider analyzing only males? What are the results if you only include males?
24. Line 398 “established experimental approach in which mice....”
25. Lines 399-401 These sentences are a little awkward and a little overstated. I think it is reasonable to say something like –“Building on a previous observational study, our experimental approach provides direct support for the hypothesis that the t haplotype affects dispersal. This is the first example of a selfish genetic element impacting dispersal and adds to a small number of known dispersal polymorphisms.”
26. Line 407- This is another example of how it would be good to spend time in the introduction laying out the previous results. The readers don’t really know here what was found before.
27. Line 407 – this paragraph could be organized a bit more to help the reader and the language could be adjusted – e.g. avoid “perhaps”, “we do not know” and just lay out what is known and possible explanations.
 - a. It would be good to lead with something like –“Evidence for increased dispersal of +/t at high densities is equivocal. Here, while we found that the fraction of +/t that dispersed vs. +/+ that dispersed was higher at higher densities, the difference was not statistically significant.....” and then list possible explanations for the differences between the studies along with the possible interpretations and significance of the results.
 - b. A power analysis could be helpful to determine if the experiment was sufficiently powered to see a difference of the same size observed in the previous study.
 - c. Were there differences in density between this study and the previous one? Age of the mice?
28. Line 421- What is the negative evidence?
29. Line 424- I am not sure what the last sentence means.
30. Line 428 “Consistent with our past observational study, we did” This paragraph needs to be tightened up. Break up the first and last sentences.
31. Line 444 – there is a typo here.
32. Line 428- Overall, the discussion of density and body size is a little disorganized and speculative. Focus on the results observed and how they generate hypotheses that could be tested in the future. I do think the body size result is interesting and merits discussion – it just needs to be more focused and directed for the reader.
33. Line 477- “In summary, the relationships between behaviors related to exploration, dispersal in the enclosure, and genotype is not yet clear. Additional experiments are needed to understand
34. Line 477 – One other possible explanation for the lack of consistency among the exploratory behavior analysis and the enclosure dispersal patterns is that mice may alter their behavior based on other mice and the environment such that tests in the lab may not map directly to results in a semi-natural enclosure with other mice.
35. Line 490 – “to be successful as immigrants”
36. Line 494 - Strike “In doing so,” replace with “Dispersal may reduce the negative fitness effects of the t haplotype”
37. Line 495 – what are its advantageous traits?
38. Line 496 – replace “makes a good case” w/ something like “provides support for the hypothesis that”

39. Line 497- "It will be interesting to see.." is a little conversational. Maybe something like "This study establishes a hypothesis that can be tested in other systems such as SR in.... Moreover, results motivate additional studies of dispersal-related behaviors in t carriers"

40. Consider adding tables as supplementary information fully reporting the details of the statistical analysis. The r scripts and results are helpful and very good to include, but having formatted tables that can be referred to in the text would help the reader interpret the results. For the supplemental data tables, please provide a key that explains column headings in the context of the experiment.

Decision letter (RSOS-202050.R0)

Dear Mr Runge

The Editors assigned to your paper RSOS-202050 "Experiments confirm a dispersive phenotype associated with a natural gene drive system" have now received comments from reviewers and would like you to revise the paper in accordance with the reviewer comments and any comments from the Editors. Please note this decision does not guarantee eventual acceptance.

Please submit your revised manuscript and required files (see below) no later than 21 days from today's (ie 02-Mar-2021) date. Note: the ScholarOne system will 'lock' if submission of the revision is attempted 21 or more days after the deadline. If you do not think you will be able to meet this deadline please contact the editorial office immediately.

on behalf of Dr Polly Campbell (Associate Editor) and Kevin Padian (Subject Editor)
 openscience@royalsociety.org

Associate Editor Comments to Author (Dr Polly Campbell):

Comments to the Author:

First, I want to apologize to the authors for the long delay in getting a decision on their manuscript. Both reviewers see value in the study and appreciate the experimental design. I agree with them and share Reviewer 2's view that this work is of potentially broad interest. I also agree that the manuscript would benefit from careful revision; both reviewers provide extensive practical suggestions that should improve clarity and increase impact. Please pay particular attention to comments on the statistical analyses (e.g., Reviewer 2, comments 21 and 27b).

Reviewer comments to Author:

Reviewer: 1

Comments to the Author(s)

See the attached document.

Reviewer: 2

Comments to the Author(s)

This study provides experimental support for the hypothesis that *t* carriers in house mice are more likely to disperse. In addition, the study includes two analyses of behaviors (in *+/+* and *+/t* mice) related to dispersal, including overall activity level and exploration. While the *t* haplotype did not contribute to differences in activity level, it did explain variation in some aspects of exploratory behavior. Understanding how the *t* haplotype influences behavior may help identify how it spreads despite significant negative effects on fitness related to sperm competition and homozygous inviability. The authors interpret their results in this context.

The experiments were generally carefully designed and the analysis clear and motivated. The experimental enclosure is unique and challenging to execute. The results have significance for both the broader research area of selfish genetic elements as well as evolutionary genetics in house mice.

Major Comments

1. Provide more context on the previous observational study to motivate this work and to lay the groundwork for interpretation of the results in the discussion.
2. The potential link between dispersal and selection against *t* carriers due to *t/t* inviability is easy for the reader to understand. The link between population density and mating multiply and selection against *t* carriers needs to be developed further in the abstract (briefly) and introduction (more room here to discuss). For example, to what extent does mating multiply (and selection against *t* carriers) depend on density? Is there a strong relationship between density and number of matings? Does the disadvantage to *t* carriers occur as a threshold (i.e. any more than one mating is very bad for *t* carriers) or is it linear (increasingly detrimental as the number of matings increase)? Does data suggest that female mice mate with at least two males in even less dense populations? Some studies are cited, but they need to be discussed.
3. There are some experimental design considerations (variation in relatedness among mice) and questions of statistical power/approach that should be addressed more directly in the text. Specific examples are given in the detailed comments.

4. There is an opportunity to strengthen this manuscript by revising the Discussion. This section is less organized and clear and the tone less formal and more speculative than the rest of the manuscript.

Detailed Comments:

1. Line 13- It would be helpful to note the type of study that produced the previous result (observational) as a contrast to this controlled, experimental approach. How strong was the evidence from the observational study?
2. Line 17- It would be good to re-phrase "heavier t-carrier" a bit. No mention of weight or body size has been made before this and it is not clear what "heavier" means. Is it weight per unit length? Is it controlled for age?
3. Line 20 – this is a little speculative. I think it is more well-supported to say that the results provide experimental evidence that the t haplotype affects dispersal and adds support to the hypothesis that dispersal may reduce the fitness costs of the t.
4. Line 33 -Explain a little bit more about genetic mechanisms of drive suppression to contrast with the work here on behavior. Readers who are less familiar with genetic drive could use more context.
5. Line 43- Give the chromosome number
6. Line 54- "Recently, in a long-term observational study on free-living house mice, we found that ...". There is also room here to elaborate on the results and the strength of the conclusions to better motivate this experimental approach.
7. Line 58- This is a good place to put in more detail about the association between density and number of mates per estrous.
8. Line 59- I think "positively" should be "positive"
9. Line 69 – Why are controlled experiments necessary? E.g. "Controlled, manipulative experiments can be used to directly test the link between the t-haplotype and dispersal at different densities."
10. Line 71 – I don't think you need this sentence here. It seems like something better for the discussion where you can bring together the results of the two studies and discuss the evidence for the hypothesis.
11. Line 74- This sentence is missing mention of the t haplotype.
12. Line 98- Can you provide some details on the genetics of the colony –i.e. how are individuals related to each other or is there a breeding scheme that can give context?
13. Line 109- Higher relatedness among the mice in the high-density treatment –i.e. males and females from the same parents seems like an experimental design problem. I appreciate that generating the mice for the experiment is very time consuming, but it is plausible that relatedness could be a confounding factor. It would be helpful to discuss the extent of the issue – e.g. if very few brother-sisters were in the same cohort than the impact could be trivial. It would also be good to talk about the natural history of house mice and what might already be known about the impact of relatedness on dispersal to help address this concern.
14. Line 121- This is subjective, but I would give the methods and results in the order of the tests. It was a bit confusing to have the enclosure and then the other two tests.
15. Line 135- were lengths measured as well?
16. Line 137 – Discuss the choice of these densities in the context of house mouse natural history rather than just the previous study.
17. Line 146 – "an average of 107.."
18. Line 154- I like the photograph, but it may be better to put this in the supplement and use a diagram here instead, so that the reader better understands the dispersal barrier as a water cage. It is difficult to see that with the plastic bin that is pretty opaque. The external cages look haphazardly set up.
19. Line 160 – I did not understand this sentence.
20. Line 162- Explain what "censored data points" are for the reader.

21. Line 166- This analysis is critical for the reader and I think it could be clearer. Explain what philopatry is in this context and why you are analyzing dispersal using a survival model type approach as opposed to alternatives. For example, you do test emergence from the tunnel later as a binary trait, did you test dispersal in the enclosures as a binary trait?
22. Line 271 - Had the mice been “cohorted” at this point?
23. Line 322- Given the very low rate of female dispersal and the low absolute number of female dispersal events, did you consider analyzing only males? What are the results if you only include males?
24. Line 398 “established experimental approach in which mice....”
25. Lines 399-401 These sentences are a little awkward and a little overstated. I think it is reasonable to say something like –“Building on a previous observational study, our experimental approach provides direct support for the hypothesis that the t haplotype affects dispersal. This is the first example of a selfish genetic element impacting dispersal and adds to a small number of known dispersal polymorphisms.”
26. Line 407- This is another example of how it would be good to spend time in the introduction laying out the previous results. The readers don’t really know here what was found before.
27. Line 407 - this paragraph could be organized a bit more to help the reader and the language could be adjusted – e.g. avoid “perhaps”, “we do not know” and just lay out what is known and possible explanations.
- a. It would be good to lead with something like –“Evidence for increased dispersal of +/t at high densities is equivocal. Here, while we found that the fraction of +/t that dispersed vs. +/+ that dispersed was higher at higher densities, the difference was not statistically significant.....” and then list possible explanations for the differences between the studies along with the possible interpretations and significance of the results.
- b. A power analysis could be helpful to determine if the experiment was sufficiently powered to see a difference of the same size observed in the previous study.
- c. Were there differences in density between this study and the previous one? Age of the mice?
28. Line 421- What is the negative evidence?
29. Line 424- I am not sure what the last sentence means.
30. Line 428 “Consistent with our past observational study, we did” This paragraph needs to be tightened up. Break up the first and last sentences.
31. Line 444 - there is a typo here.
32. Line 428- Overall, the discussion of density and body size is a little disorganized and speculative. Focus on the results observed and how they generate hypotheses that could be tested in the future. I do think the body size result is interesting and merits discussion – it just needs to be more focused and directed for the reader.
33. Line 477- “In summary, the relationships between behaviors related to exploration, dispersal in the enclosure, and genotype is not yet clear. Additional experiments are needed to understand
34. Line 477 - One other possible explanation for the lack of consistency among the exploratory behavior analysis and the enclosure dispersal patterns is that mice may alter their behavior based on other mice and the environment such that tests in the lab may not map directly to results in a semi-natural enclosure with other mice.
35. Line 490 – “to be successful as immigrants”
36. Line 494 - Strike “In doing so,” replace with “Dispersal may reduce the negative fitness effects of the t haplotype”
37. Line 495 – what are its advantageous traits?
38. Line 496 - replace “makes a good case” w/ something like “provides support for the hypothesis that”
39. Line 497- “It will be interesting to see..” is a little conversational. Maybe something like “This study establishes a hypothesis that can be tested in other systems such as SR in.... Moreover, results motivate additional studies of dispersal-related behaviors in t carriers”

40. Consider adding tables as supplementary information fully reporting the details of the statistical analysis. The r scripts and results are helpful and very good to include, but having formatted tables that can be referred to in the text would help the reader interpret the results. For the supplemental data tables, please provide a key that explains column headings in the context of the experiment.

===PREPARING YOUR MANUSCRIPT===

===PREPARING YOUR REVISION IN SCHOLARONE===

<https://royalsociety.org/journals/authors/author-guidelines/#supplementary-material> to include a suitable title and informative caption. An example of appropriate titling and captioning may be found at https://figshare.com/articles/Table_S2_from_Is_there_a_trade-off_between_peak_performance_and_performance_breadth_across_temperatures_for_aerobic_sc_ope_in_teleost_fishes_/3843624.

Author's Response to Decision Letter for (RSOS-202050.R0)

See Appendix B.

Decision letter (RSOS-202050.R1)

Dear Mr Runge,

It is a pleasure to accept your manuscript entitled "Experiments confirm a dispersive phenotype associated with a natural gene drive system" in its current form for publication in Royal Society Open Science. The comments of the reviewer(s) who reviewed your manuscript are included at the foot of this letter.

on behalf of Dr Polly Campbell (Associate Editor) and Kevin Padian (Subject Editor)
openscience@royalsociety.org

Subject Editor Comments to Author (Professor Kevin Padian):
Comments to the Author:

Thanks for your revisions. The AE is pleased with your efforts and we are happy to accept your manuscript. Best wishes.

Appendix A

Review of ms. RSOS-202050 Runge & Lindholm: Experiments confirm a dispersive phenotype associated with a natural gene drive system

The paper reports on results of a follow-up study complementing the work published in Proc. R. Soc. B 285 (2018). The authors conclude $+/t$ mice are more dispersive but without any dependence on population density. Heavier-than-average t -carriers were more likely to disperse than heavier-than-average wild type mice. Moreover, $+/t$ individuals were also found to be more explorative. By contrast, the study found no differences in activity. The authors discuss possible reasons for differences between the recent and previous findings obtained by the same group.

The study is well designed, however, I can't help thinking that some of the conclusions are not firmly grounded in the data. In particular, I don't see any sound evidence the two genotypes differ in "explorativeness" (summarized through PC1; figure 6). The only significantly different behavioural trait under study was dispersal. But it should be noted that only 24 mice (mostly males) emigrated from the enclosure and as such were subsequently tested. One may wonder whether low sample sizes can be responsible for the lack of significance in some of the tests. On the other hand, sometimes the analyses resembled a fishing expedition with many parameters and their interactions tested: the more tests, the more potential false positives (although these were not, strictly speaking, multiple pairwise tests, perhaps some kind of adjustment of the alpha value should be applied).

The only predictor of future propensity to disperse was whether mice emerged from tunnels, with marginal significance (lines 391-392). Unfortunately, it is not clear what was the proportion of $+/t$ individuals out of the 51 mice that did emerge and those 59 that did not.

The text is sometimes hard to follow and I encourage the authors to rephrase some sentences (see below). The description of the exploration experiment is not clear: no division of the setup (Fig. 2) into compartments is mentioned in the text. On line 252, the authors say the maximum number of visited compartments to be 5. However, I can see 9 (left) + 9 (right) + at least 8 (central cage) unoccupied compartments in Fig. 2. So it seems the mice were allowed only to visit the central cage. If so, this should be described in more detail; if not, the compartmentation should be described more precisely. How is a visit of the n -th compartment defined? When a mouse's snout appears beyond the demarcating line? Or paws of one of his/her forelegs? Both forelegs, all four legs?

Other comments are below:

Line 59: limited potential -> a limited potential

L60: to successfully sire offspring -> to sire offspring successfully

L70: are however -> are, however,

L89: replicate enclosure -> a replicate enclosure

L111: the suitable -> suitable

L164: fate, -> fate

L197: something seems to be missing in the sentence ($+/+$ female?)

L201: analysis -> the analysis

L207: "per hour-of-day averages" should be rephrased to make the sentence more clear

L210: based on visual inspection of the revolutions per hour of day -> based on the visual inspection

L220: the phrase "many more less active" is unclear

L233: touching the mice -> touching the mouse

L246: to not explore -> not to explore

L258-259: the sentence "To do so, we set the times at which a mouse entered the n th compartment for mice that did not enter n compartments to 25 minutes..." should be re-formulated, it is hard to follow

L264: outside of the tunnel -> outside the tunnel

L281: generalized linear model -> a generalized linear model

L335: producing 372.45 -> producing by 372.45 ?

L347-348 and Methods: "to avoid multiple testing of the strongly correlated variables" -> by using PCA, you *do not avoid* multiple testing!

Figure 7 is referred to before Figure 6

L359-361: The sentence could be, perhaps, a bit rephrased; it is not clear what "an increased PC1" means – increased relative to what? What does the value 1.36 mean?

L369-370: Consequently, genotype did not explain whether they did so better than no predictors ... it is not clear what "they" and "no predictors" is referring to

L431: Surprisingly however, -> Surprisingly, however,

Appendix B

Dear Polly Campbell,

thank you for the decision and organising the reviews. We also thank the reviewers for their highly valuable and helpful feedback. Below, we will respond to the comments made.

We will quote reviewers with a leading “>” and in a monospaced font and then respond below. We have grouped some reviewer points when they belonged together or when they contained small changes. The changes can also be seen in the tracked changes document.

Reviewer 1

> In particular, I don't see any sound evidence the two genotypes differ in “explorativeness”

We defined “explorativeness” in our analyses as the first dimension of the PCA that is made up of the behaviours of the mice that emerged from the tunnel and began exploring. We did so because it is highly correlated with all of these variables, such as time until emergence or number of compartments explored. We provided statistical evidence that this definition of “explorativeness” differs between +/+ and +/t. We assume the reviewer may refer to the fact that the genotypes did not differ in their probability to emerge from the tunnel and begin exploring. We agree that this is to some extent contrary evidence, but similar behaviours were previously interpreted as measures of anxiety rather than exploration. We have added this and also changed the text to reflect the level of certainty better.

> But it should be noted that only 24 mice (mostly males) emigrated from the enclosure and as such were subsequently tested.

We are addressing this feedback by clarifying that the Cox models of dispersal are incorporating data on all 135 mice, dispersers and non-dispersers, and as such the sample size is 135, not 24.

> One may wonder whether low sample sizes can be responsible for the lack of significance in some of the tests.

We have added a power analysis for the dispersal models (see response to reviewer 2 on this issue). For the activity tests, sample size should not be an issue given that the wheel revolutions of the genotypes were very similar, thus we do not expect any changes to the results with increased sample sizes. The explorativeness analysis did show a significant difference, hence we assume that this was not the one the reviewer referred to.

> On the other hand, sometimes the analyses resembled a fishing expedition with many parameters and their interactions tested: the more tests, the more potential false positives

We agree that there are many tests, but we have separated hypothesized from not hypothesized tests which we are now pointing out more clearly.

> Unfortunately, it is not clear what was the proportion of +/t individuals out of the 51 mice that did emerge and those 59 that did not

Indeed, this information was not presented clearly enough, which we have now improved.

> The description of the exploration experiment is not clear: no division of the setup (Fig. 2)

into compartments is mentioned in the text. [...] How is a visit of the n-th compartment defined?

We apologize for the confusion that resulted from the photograph of the experiment. The experiment does consist of five compartments, three cages and two tubes. The lines drawn on the cages that can be seen in the photograph were not used for analysis in our study. The photograph has been moved to the supplementary and a schematic takes its place. We have also clarified the text with regards to what constitutes a compartment and when a mouse was considered as having entered one.

- > Line 59: limited potential -> a limited potential
- > L60: to successfully sire offspring -> to sire offspring successfully
- > L70: are however -> are, however,
- > L89: replicate enclosure -> a replicate enclosure
- > L111: the suitable -> suitable
- > L164: fate, -> fate
- > L197: something seems to be missing in the sentence (+/+ female?)
- > L201: analysis -> the analysis
- > L207: "per hour-of-day averages" should be rephrased to make the sentence more clear
- > L210: based on visual inspection of the revolutions per hour of day -> based on the visual inspection
- > L220: the phrase "many more less active" is unclear
- > L233: touching the mice -> touching the mouse
- > L246: to not explore -> not to explore
- > L258-259: the sentence "To do so, we set the times at which a mouse entered the nth compartment for mice that did not enter n compartments to 25 minutes..." should be re-formulated, it is hard to follow
- > L264: outside of the tunnel -> outside the tunnel
- > L281: generalized linear model -> a generalized linear model
- > L335: producing 372.45 -> producing by 372.45 ?
- > L347-348 and Methods: "to avoid multiple testing of the strongly correlated variables" -> by using PCA, you do not avoid multiple testing!
- > Figure 7 is referred to before Figure 6
- > L369-370: Consequently, genotype did not explain whether they did so better than no predictors ... it is not clear what "they" and "no predictors" is referring to
- > L431: Surprisingly however, -> Surprisingly, however,

We agree and have made adjustments to the text.

- > L359-361: The sentence could be, perhaps, a bit rephrased; it is not clear what "an increased PC1" means - increased relative to what? What does the value 1.36 mean?

We are now translating this change in PC1 by providing two examples of the underlying variables and how much they change with such an increase in PC1.

Reviewer 2

- > 1. Provide more context on the previous observational study to motivate this work and to lay the groundwork for interpretation of the results in the discussion.
- > 1. Line 13- It would be helpful to note the type of study that produced the previous result (observational) as a contrast to this controlled, experimental approach. How strong was the evidence from the observational study?

> 6. Line 54- "Recently, in a long-term observational study on free-living house mice, we found that". There is also room here to elaborate on the results and the strength of the conclusions to better motivate this experimental approach.

> 26. Line 407- This is another example of how it would be good to spend time in the introduction laying out the previous results. The readers don't really know here what was found before.

We have now mentioned the type of the previous study in the abstract and expanded the introductory paragraph to more fully lay out the previous results.

> The link between population density and mating multiply and selection against t carriers needs to be developed further in the abstract (briefly) and introduction (more room here to discuss).

> 7. Line 58- This is a good place to put in more detail about the association between density and number of mates per estrous.

We are now summarising the referenced studies to point out this relationship better.

> There are some experimental design considerations (variation in relatedness among mice) and questions of statistical power/approach that should be addressed more directly in the text.

> 13. Line 109- Higher relatedness among the mice in the high-density treatment—i.e. males and females from the same parents seems like an experimental design problem. I appreciate that generating the mice for the experiment is very time consuming, but it is plausible that relatedness could be a confounding factor. It would be helpful to discuss the extent of the issue—e.g. if very few brother-sisters were in the same cohort than the impact could be trivial. It would also be good to talk about the natural history of house mice and what might already be known about the impact of relatedness on dispersal to help address this concern.

We are now clarifying that we did always use eight brother-sister pairs in the high density. Our previous analysis of the observational study also included a supplementary showing that relatedness does decrease dispersal, but without affecting the genotype's effect. We mention this now in the text as well.

> 2. Line 17- It would be good to re-phrase "heavier t-carrier" a bit. No mention of weight or body size has been made before this and it is not clear what "heavier" means. Is it weight per unit length? Is it controlled for age?

Agreed, we changed it to "above-average body weight". This is not directly controlled for age, because age (and sex for that matter) would be controlled for in the models if the terms are included, but age was never an important predictor in the models.

> 3. Line 20 - this is a little speculative. I think it is more well-supported to say that the results provide experimental evidence that the t haplotype affects dispersal and adds support to the hypothesis that dispersal may reduce the fitness costs of the t.

We have changed the language accordingly.

> 4. Line 33 -Explain a little bit more about genetic mechanisms of drive suppression to contrast with the work here on behavior. Readers who are less familiar with genetic drive could use more context.

We have added an example.

- > 5. Line 43- Give the chromosome number
- > 8. Line 59- I think "positively" should be "positive"

Done.

- > 9. Line 69 - Why are controlled experiments necessary? E.g. "Controlled, manipulative experiments can be used to directly test the link between the t-haplotype and dispersal at different densities."

Agreed and adjusted.

- > 10. Line 71 - I don't think you need this sentence here.
- > 11. Line 74- This sentence is missing mention of the t haplotype.

Done.

- > 12. Line 98- Can you provide some details on the genetics of the colony -i.e. how are individuals related to each other or is there a breeding scheme that can give context?

All individuals are related in so far as they all genetically originate from the same source population, but with slightly different generations between them. As we mentioned in the text, the mice are offspring of a lab colony that was founded by mice from the long-term wild study, and the colony is regularly 'refreshed' with newly caught mice from the field study.

We have added to the text that 12 individual parents of the 34 breeding pairs were caught from the field study while the rest was born in the lab (F1-F3 lab generations from the long-term study).

- > 14. Line 121- This is subjective, but I would give the methods and results in the order of the tests. It was a bit confusing to have the enclosure and then the other two tests.

We have debated this. Dispersal is the main result and topic of the study, which is why we chose this order. The feedback as a whole is driving us in different directions: Reviewer 1 is less positive about the other results (explorativeness), while reviewer 2 suggests a change of the order. We propose that we keep the order as it is in light of all things considered.

- > 15. Line 135- were lengths measured as well?

No, lengths were not measured.

- > 16. Line 137 - Discuss the choice of these densities in the context of house mouse natural history rather than just the previous study.

In the methods, we have now a detailed comparison to the previous study. In the discussion, we are now also comparing the densities to the range of mouse densities found in the wild.

- > 17. Line 146 - "an average of 107.."

Adjusted.

- > 18. Line 154- I like the photograph, but it may be better to put this in the supplement and use a diagram here instead, so that the reader better understands

the dispersal barrier as a water cage. It is difficult to see that with the plastic bin that is pretty opaque. The external cages look haphazardly set up.

The photograph is now in the SI and has been replaced with a schematic of the setup. The dispersal cages may look quite improvised, but caused no issues.

> 19. Line 160 - I did not understand this sentence.

We have rewritten and split the sentence.

> 20. Line 162- Explain what "censored data points" are for the reader.

Done.

> 21. Line 166- This analysis is critical for the reader and I think it could be clearer. Explain what philopatry is in this context and why you are analyzing dispersal using a survival model type approach as opposed to alternatives. For example, you do test emergence from the tunnel later as a binary trait, did you test dispersal in the enclosures as a binary trait?

We are now pointing out why we think the analysis is better this way, but we also include a binary trait version in the SI, but we only mention, not discuss it, in the manuscript. In short, results are overall similar with this approach, but the p-value of adding genotype to the model is slightly increased and no longer below 0.05 (0.063 and 0.055 depending on the comparison).

> 22. Line 271 - Had the mice been "cohorted" at this point?

At this point, the mice of a cohort had in common that they were transferred out of the breeding lab at the same time and had been in a separate room together for a few days, although each in their own cage. By testing for the effect, we want to ensure that there is no influence of this that we miss. Thankfully, this was not the case. We have added a half sentence to clarify our intention.

> 23. Line 322- Given the very low rate of female dispersal and the low absolute number of female dispersal events, did you consider analyzing only males? What are the results if you only include males?

Given our previous results, we thought it correct to analyse males and females combined, while controlling/testing for an influence of sex. We can see how a sex-separated analysis can be of interest and have included male-only models in the SI. In short, in these analyses the effect of the genotype is much more readily observed, but the genotype x weight interaction is no longer significant. We present histograms of body weight for males and females, divided by genotype, and conclude that the change in significance is not due to female (non-)dispersers driving it, but more likely due to the reduction in sample size when only considering males.

> 24. Line 398 "established experimental approach in which mice...."
Adjusted.

> 25. Lines 399-401 These sentences are a little awkward and a little overstated. I think it is reasonable to say something like -"Building on a previous observational study, our experimental approach provides direct support for the hypothesis that the t haplotype affects dispersal. This is the first example of a selfish genetic element impacting dispersal and adds to a small number of known dispersal polymorphisms."

Rephrased.

> 27. Line 407 - this paragraph could be organized a bit more to help the reader and the language could be adjusted—e.g. avoid “perhaps”, “we do not know” and just lay out what is known and possible explanations. a. It would be good to lead with something like—“Evidence for increased dispersal of +/t at high densities is equivocal. Here, while we found that the fraction of +/t that dispersed vs. +/+ that dispersed was higher at higher densities, the difference was not statistically significant.....” and then list possible explanations for the differences between the studies along with the possible interpretations and significance of the results.

We have rewritten the paragraph to accommodate this helpful feedback.

> b. A power analysis could be helpful to determine if the experiment was sufficiently powered to see a difference of the same size observed in the previous study.

Such an analysis has been included alongside the binary trait analysis in the SI. We conclude that the power was 88.4% to detect the genotype x density interaction and 67.7% to detect the genotype effect. This is calculated by considering only the sample size of founders, but we originally intended, as stated in the manuscript, to test for dispersal in the offspring, too, but their rate of dispersal was surprisingly much too low. With such a much higher sample size, the power would have been even higher.

> c. Were there differences in density between this study and the previous one? Age of the mice?

We are now more deeply comparing the two studies.

> 28. Line 421- What is the negative evidence?

> 29. Line 424- I am not sure what the last sentence means.

We have rephrased these sentences to make them clearer.

> 30. Line 428 “Consistent with our past observational study, we did” This paragraph needs to be tightened up. Break up the first and last sentences.

We have incorporated the feedback and also broken this paragraph up into the two main themes.

> 31. Line 444 - there is a typo here.

We have rewritten this part.

> 32. Line 428- Overall, the discussion of density and body size is a little disorganized and speculative. Focus on the results observed and how they generate hypotheses that could be tested in the future. I do think the body size result is interesting and merits discussion—it just needs to be more focused and directed for the reader.

We have rewritten large parts of these sections with the reviewer’s feedback in mind.

> 33. Line 477- “In summary, the relationships between behaviors related to exploration, dispersal in the enclosure, and genotype is not yet clear. Additional experiments are needed to understand”

Adjusted.

> 34. Line 477 - One other possible explanation for the lack of consistency among the exploratory behavior analysis and the enclosure dispersal patterns is that mice may alter their behavior based on other mice and the environment such that tests in the lab may not map directly to results in a semi-natural enclosure with other mice.

We have added this potential explanation with suggestions for future experiments.

> 35. Line 490-"to be successful as immigrants"

> 36. Line 494 - Strike "In doing so," replace with "Dispersal may reduce the negative fitness effects of the t haplotype"

> 37. Line 495-what are its advantageous traits?

> 38. Line 496 - replace "makes a good case" w/ something like "provides support for the hypothesis that"

> 39. Line 497- "It will be interesting to see.." is a little conversational. Maybe something like "This study establishes a hypothesis that can be tested in other systems such as SR in.... Moreover, results motivate additional studies of dispersal-related behaviors in t carriers"

Agreed and adjusted.

> 40. Consider adding tables as supplementary information fully reporting the details of the statistical analysis. The r scripts and results are helpful and very good to include, but having formatted tables that can be referred to in the text would help the reader interpret the results. For the supplemental data tables, please provide a key that explains column headings in the context of the experiment.

We have added supplementary information containing 1) the photos that were previously in the manuscript, 2) binary trait dispersal analyses, 3) Males only dispersal analyses, and 4) tabular model outputs. We have also added a key to the data files that are uploaded to Zenodo as written in the manuscript's data availability section.